# Can flavoprotein monooxygenases functionalize long-chain *n*-alkanes?

Raul Mireles[1]*, Arne Matthews[2], Robin Teufel[2], Lianet Noda-García [1]*

1 Institute of Environmental Sciences, The Robert H. Smith Faculty of Agriculture, Food and Environment, The Hebrew University of Jerusalem, Rehovot, Israel, 2 Pharmaceutical Biology, Department of Pharmaceutical Sciences, University of Basel, Basel, Switzerland

* raul.mireles@mail.huji.ac.il (RM); lianet.noda@mail.huji.ac.il (LN-G)

## Abstract

Long-chain *n*-alkane functionalization is relevant for industrial and environmental applications, but it remains a significant challenge due to the inherent stability of their covalent bonds. Biocatalytic approaches offer promising strategies due to their potential for selective, efficient, and environmentally friendly processes. Among the enzyme families known to functionalize long-chain *n*-alkanes, one iron-binding protein, AlkB, has been characterized. Additionally, two distinct metal-free flavin-dependent enzymes, LadA and AlmA, are presumed to perform this function. Unlike the membrane-bound AlkB, LadA and AlmA are soluble proteins, making them more amenable to engineering and scalable industrial applications. In this study, we attempted to reproduce and optimize the *n*-alkane monooxygenase activity of LadA. We tested the functionality of this enzyme, an optimized variant, and four novel homologs under *in vitro* conditions. Despite extensive efforts, we were unable to detect any long-chain *n*-alkane hydroxylation. Analysis of LadA's protein superfamily and the reported experimental evidence indicate that LadA may be involved in the metabolism of long-chain *n*-alkanes. However, its role in the first oxo-functionalization step could not be corroborated. Similar conclusions were published regarding AlmA's activity. Altogether, these findings challenge the current understanding of flavoprotein monooxygenases as long-chain *n*-alkane monooxygenases and underscore the need for further investigation into their biochemical role.

## Introduction

Aliphatic alkanes (*n*-alkanes) are ubiquitous and abundant hydrocarbons in diverse ecosystems [1]. In an industrial context, they represent the fuel potential of crude oil, as they constitute up to 30% of its total weight [2,3]. These molecules can be classified based on their chain length, which directly influences their physical and chemical properties, as well as their practical applications [4]. Short- and medium-chain *n*-alkanes, ranging from one to sixteen carbon atoms, are primarily found in gaseous

**Data availability statement:** All relevant data are within the manuscript and its Supporting Information files.

**Funding:** Proposal no. 12120006 Ministry of Agriculture and Rural Development. The funders had no role in study design, data collection and analysis, decision to publish, or preparation of the manuscript.

**Competing interests:** The authors have declared that no competing interest exist.

and liquid forms. Due to their high combustibility, short and medium-chain n-alkanes are mainly used as fuel for power generation. On the other hand, long-chain *n*-alkane (17 + carbons) exist as solid compounds at room temperature and are commonly referred to as paraffin waxes. The physical state and extremely low reactivity of long-chain *n*-alkanes have prevented them from being extensively used in industrial applications. Hence, they are often considered contaminants in the oil industry and environment [5,6].

Processes capable of transforming long-chain *n*-alkanes for repurposing them for human use have been pursued for decades [7–9]. Despite advancements, challenges persist due to the chemical catalysts' limited efficiency, selectivity, cost, and energy-demanding conditions [10]. Along those lines, microorganisms represent a vast reservoir of biocatalysts, providing enzymes that offer novel solutions to such challenging reactions. Indeed, several bacterial species have been described to utilize solid *n*-alkanes as the sole carbon and energy source, showing that these transformations are feasible under physiological conditions, which are milder than the high temperatures or harsh solvents needed for chemical processes [11]. Members of three bacterial enzyme families were proposed to catalyze the terminal monooxygenation of long-chain *n*-alkanes: the alkane 1-monooxygenase, AlkB, belonging to a family of iron-binding transmembrane proteins [12,13], and members of two metal-free protein families, alkane metabolism A (AlmA), a group B flavoprotein monooxygenase (FPMO) having a Rossman-like protein fold, and long-chain alkane degradation (LadA) with a TIM-barrel-like fold, classified as group C FPMO [14–16]. Unlike AlkB, the FPMOs AlmA and LadA are soluble proteins amenable to engineering, making them promising targets for biocatalyst improvement.

Initial studies on the *in vivo* function of AlmA in *Acinetobacter sp.* DSM 17874 highlighted its crucial role in long-chain *n*-alkane bacterial metabolism. A mutant strain with a disrupted *almA* gene could not assimilate long-chain *n*-alkanes [15]. Although the specific function of AlmA was not detailed in the original publication, it was subsequently interpreted as an *n*-alkane monooxygenase [16–18]. Later, an AlmA homolog in *Alcanivorax dieselolei* B5 (50% sequence identity to DSM 17874) was purified and biochemically characterized as a long-chain *n*-alkane monooxygenase, reinforcing this interpretation. Recently, however, a close homolog of AlmA in *Acinetobacter baylyi* ADP1 (77% sequence identity to DSM 17874) was discovered to function not as a long-chain *n*-alkane but as a Baeyer-Villiger monooxygenase (BVMO). Thus, while *in vivo* studies unambiguously show AlmA is involved in long-chain *n*-alkane metabolism, *in vitro* experiments revealed that its activity could target pathway intermediates such as already activated and oxidized long-chain aliphatic ketones rather than long-chain *n*-alkanes [19].

With AlmA's controversial activity, LadA appeared to remain the only soluble and engineerable enzyme candidate for long-chain *n*-alkane functionalization. Thus, we initiated investigations to deepen our understanding of its structure and function relationship and explore its applications. However, despite successfully purifying different LadA enzymes, our teams -the Noda-García and the Teufel groups- independently, were unable to replicate the findings from three studies from the same group that

reported LadA's long-chain *n*-alkane monooxygenase activity [14,20,21]. Our work thoroughly examines reaction conditions and the biocatalyst behavior, leading to a reevaluation of LadA's potential function. Together with AlmA's data, we propose a reassessment of metal-free, flavin-dependent enzymes in functionalizing medium to long-chain *n*-alkanes.

## Results

### 1. The optimization of reaction conditions is ineffective in achieving the reported LadA activity

LadA belongs to the group C FPMO family [22]. It is the monooxygenase component of a two-enzyme system that requires molecular oxygen ($O_2$) and the reduction of its flavin mononucleotide (FMN) cofactor. Like other group members, LadA cannot reduce FMN and therefore relies on the activity of an independent enzyme -a FMN reductase- that reduces FMN to $FMNH_2$ using NAD(P)H as a cofactor. The $FMNH_2$ is then transferred to the monooxygenase component via protein-protein interaction or free diffusion [23]. Then, the $FMNH_2$ reacts with $O_2$ to produce the catalytic intermediate, a flavin hydroperoxy species, responsible for the oxygenation of the organic substrate. Therefore, the monooxygenase and reductase components are essential for the reaction, along with NAD(P)H and FMN as cofactors.

Three previous studies from the same research group have reported the biochemical activity of one LadA enzyme [14,20,21]. The activity was assessed by detecting the hydroxylated product of hexadecane (C16), 1-hexadecanol, using gas chromatography-mass spectrometry (GC-MS). In the study by Feng et al. (2007), the reaction setup included 1 mM hexadecane and 1 mM $MgSO_4$ in a 50 mM Tris-HCl buffer at pH 7.5, along with 1 mM NADH. The second study by Li et al. (2008), which focused on the structural analysis of LadA, also used 1 mM hexadecane in a 50 mM Tris-HCl buffer at pH 7.5; however, NADH was omitted, and 1 mM $FMNH_2$ was added instead. Finally, the study by Dong et al. (2012) prepared the reactions with 1 mM FMN and 10 mM NADPH in a 50 mM Tris-HCl buffer at a pH of 8.0. This study also employed a commercial surfactant, Plysurf at 0.015% (m/v), to facilitate hexadecane solubilization. In the three studies, details on the enzyme concentration used were not provided, and none of the studies mentioned adding any source for generating $FMNH_2$.

We successfully expressed and purified the same LadA protein and attempted to replicate the previously reported assay conditions, as they were ambiguously described and lacked a clear biochemical rationale (S1 Fig). Most notably, the omission of a flavin reductase despite including FMN and NADH.

We then focused on establishing more mechanistically grounded conditions. To this end, we generated $FMNH_2$ *in situ* by supplementing the reaction with purified *E. coli* flavin reductase (Fre), maintaining a 1:3 ratio relative to LadA (Fig 1A). This setup was successfully tested for other group C FPMOs [23–25]. The stability of Fre at elevated temperatures is

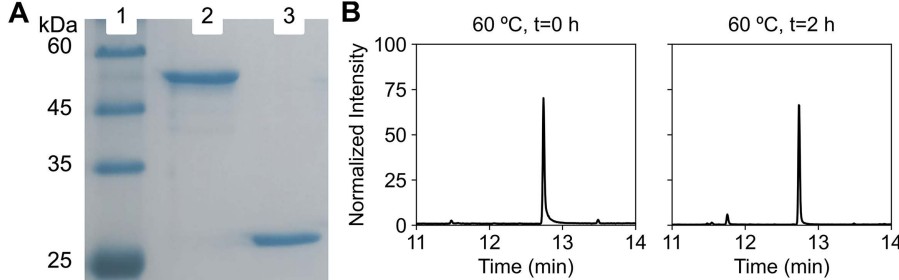

**Fig 1. Protein expression, purification, and GC-MS analyses for LadA and Fre reaction.** (A) SDS-PAGE. Lane 1: Molecular weight (MW) marker. Lane 2: purified *wild-type* N-strepTag LadA (WP_011888513.1, estimated MW = 51.8 kDa). Lane 3: purified *wild-type* N-strepTag Fre (WP_000209826.1, estimated MW = 27.5 kDa). (B) The gas chromatogram showed no hexadecane ($t_R$ = 12.74 min) conversion after two hours of incubation at 60°C. Reaction conditions were *wild-type* N-strepTag LadA (18 µM), *wild-type* N-strepTag Fre (6 µM), hexadecane (1 mM), FMN (1 mM), NADH (1 mM) and HITENOL-AR 0.5% (m/v). kDa - kilodaltons. Additional peaks at $t_R$ = 11.54 min and $t_R$ = 11.84 min correspond to pentadecane (internal standard) and butylated hydroxytoluene (contamination), respectively.

undocumented; therefore, we assessed monooxygenase activity at both 37°C, the optimal temperature for Fre, and 60°C, the reported optimal temperature for LadA. [2] Unfortunately, the three previous studies on LadA did not report specific protein concentrations. Other group C FPMOs were typically evaluated at concentrations ranging from 0.01 µM to 90 µM. Thus, we added approximately 1 g/L (18 µM) LadA to our assays, which represented the highest possible concentration before protein aggregation occurred [20,26]. [3] The surfactant Plysurf A210G (octyl phenol ethoxylate) is no longer distributed. Consequently, we employed alternative anionic surfactants for hexadecane solubilization. We used HITENOL-AR10 (DSK, Japan), a surfactant produced by the same company that manufactured the Plysurf A210G, which presumably has similar properties (personal communication with DSK). [4] Finally, while the three previous studies added $Mg^{2+}$, we omitted it. LadA lacks metal-binding sites, and no metal dependency has been reported for group C FPMOs [20]. Under these new conditions, we evaluated LadA's long-chain n-alkane monooxygenase activity. Despite extensive optimization, no hydroxylated products or hexadecane consumption were observed (Fig 1B). Detailed conditions and GC-MS analyses are presented in the methods section and supplementary information (S1 Fig).

## 2. One optimized LadA and four homologs are ineffective in yielding the activity

We hypothesized that the lack of activity could be related to the enzyme's instability under such denaturing reaction conditions. So, we enhanced LadA's robustness using the Protein Repair One-Stop Shop (PROSS) algorithm. This method incorporates atomistic modeling and phylogenetic data to design sequence variants with enhanced stability and solubility [27]. We assessed the expression, purification, and activity of two PROSS variants: LadA:P2 with eight amino acid substitutions, which did not express, and LadA:P5 with 18 amino acid substitutions, which showed a 3-fold increase in expression (S1 Table). We tested LadA:P5 in the optimized assay conditions. However, it was also inactive (S2 Fig).

Next, we tested whether the activity could be detected in other LadA homologs. We picked four LadA proteins with reported activity: the LadA from *G. thermodenitrificans* NG80−2, the only homologue with *biochemical* evidence of activity, and three LadA homologs from *G. thermoleovorans* B23 (LadB, LadAα, and LadAβ) for which the evidence was microbiological. These three homologues enhanced the growth of *Pseudomonas fluorescens* KOB2Δ1 on n-alkanes. *P. fluorescens* KOB2Δ1 lacks the AlkB enzyme, hence it grows significantly slower than the wildtype on n-alkanes as a carbon source [28,29]. The four protein sequences were used as queries to search across 215,713 bacterial proteomes in the NCBI RefSeq database (August 2021). This search yielded 660 homologs with >50% global sequence identity and >70% coverage to at least one of the query sequences. We then organized these sequence homologs into a maximum-likelihood cladogram and divided it visually into eight clades (Fig 2A). We generated a consensus sequence (cutoff >50%) for each clade by aligning all corresponding sequences. From each clade, we then selected the closest similar sequence to the consensus sequence (Fig 2B, S2 Table), which is likely to result in increased structural and functional stability of the respective proteins [30]. Further, we verified that the chosen sequences contained the conserved FTNVAF motif previously proposed to be important for interaction with the FMN cofactor and modulation of its oxygen-reactivity [31,32]. The presence of this motif thus suggests similar flavin binding features and catalytic properties to those in the query sequences. We then successfully expressed and purified four out of the eight representative sequences (S3 Fig). None of these proteins, however, demonstrated detectable long-chain *n*-alkane monooxygenase functionality under our assay conditions (S4 Fig).

## 3. Exploring other plausible roles for LadA in hydrocarbon metabolism

The available microbiology and genetic data suggest that LadA activity is indeed related to medium- and long-chain *n*-alkane metabolism: (1) *ladA*'s gene transcript was upregulated when culturing *G. thermodenitrificans* NG80−2 in crude oil, a complex mixture of hydrocarbons, including medium- and long-chain *n*-alkanes; (2) In a proteomic analysis, the LadA protein from the same organisms was overexpressed when growing in hexadecane as the carbon source compared to sucrose. (3) The heterologous expression of LadA in *Pseudomonas fluorescens* KOB2Δ1 enhanced its growth

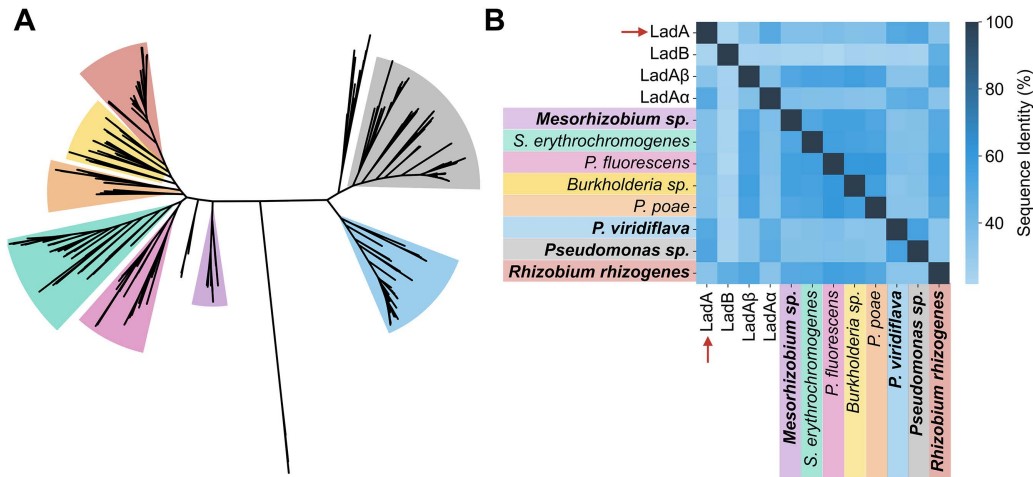

**Fig 2. LadA sequence homologs.** (A) Cladogram displaying the organization of 660 putative LadA homologs. (B) Pairwise global sequence identity comparison of LadA query sequences and the eight homologs we tested. The red arrows indicate the only biochemically characterized sequence, *wild-type* LadA (WP_011888513.1). The homologues we managed to express, purify, and perform biochemical tests are shown in bold letters.

on *n*-alkanes. *P. fluorescens* KOB2Δ1, lacking the AlkB gene, grows more slowly than the wild type on *n*-alkanes as the carbon source. The fact that LadA enhanced the growth rate could indicate that this enzyme performs the same function as AlkB but could also be reducing a bottleneck in a downstream metabolic step [12]. Since we did not detect *n*-alkane hydroxylation, we attempted to test other hypotheses connecting LadA activity to *n*-alkane or oil-related hydrocarbon metabolism outside the first step in metabolism, the hydroxylation.

A similar observation was recently reported for the enzyme family AlmA, which was initially thought to hydroxylate *n*-alkanes but was later recognized as acting downstream on the metabolic pathway by converting aliphatic ketones to esters *via* a Baeyer-Villiger monooxygenation (BVMO) [19]. BVMO activity has been reported for three other group C FPMO members, like LadA. Among them, 2,5-diketocamphane monooxygenase (2,5-DKCMO, no significant sequence identity to LadA) and 3,6-diketocamphane monooxygenase (3,6-DKCMO, 25% sequence identity to LadA), both related to camphor metabolism, and catalyzing the insertion of an oxygen atom adjacent to the carbonyl group of 2,5- and 3,6-diketocamphane, respectively [33]. Additionally, luciferases (LuxAB, 25–37% sequence identity to LadA) from *Photobacterium phosphoreum* NCIMB 844 demonstrated activity on aliphatic, monocyclic, and bicyclic ketones [34]. Thus, we tested if LadA could show BVMO activity on 2-hexadecanone as the substrate. However, we could not detect ester formation or 2-hexadecanone depletion (S5 Fig).

Moreover, a functionally characterized relative of LadA is the group C FPMO DszA from *Rhodococcus erythropolis* D1 (35% sequence identity to LadA) that functions as dibenzothiophene sulfone monooxygenase [35]. Since transcriptional analysis of the *ladA* gene showed overexpression while exposing the bacteria to crude oil containing sulfur compounds, we hypothesized that LadA might be active on dibenzothiophene sulfone, the most abundant sulfur compound in crude oil mixtures [2]. However, we could not detect any LadA activity on dibenzothiophene sulfone under the conditions reported for DszA (S6 Fig). The specific role of LadA in long-chain *n*-alkane or hydrocarbon metabolism, therefore, remains to be elucidated.

## Discussion

Group C FPMOs, such as LadA, catalyze oxygenation reactions across a diverse range of organic compounds, including those containing carbonyl (C=O), carbon-sulfur (C-S), and carbon-nitrogen (C-N) bonds. The selectivity of these

transformations is partly dictated by the nature of the oxygen-activated flavin intermediates that form during catalysis [31]. For instance, LuxAB catalyze the insertion of oxygen into aliphatic aldehydes and ketones via the formation of a C4a-hydroperoxyflavin ($Fl_{C4aOO}$) intermediate, where the oxygen is activated at the C4a position of the isoalloxazine ring in FMN (Fig 3A). Conversely, DszA most likely catalyzes the oxygenation of dibenzothiophene sulfone using a distinct intermediate, the N5-peroxyflavin ($Fl_{N5OO}$), in which the oxygen is activated at the N5 position of the flavin moiety (Fig 3B) [36]. Despite their mechanistic differences, both intermediates initiate oxygen insertion through a nucleophilic attack on an electrophilic carbon atom in the organic substrate. Thus, the chemical environment surrounding this carbon atom is a key determinant of substrate susceptibility to oxygenation, shaping the selectivity and scope of this enzyme family [37].

    n-Alkanes are molecules with an absence of a strong bond polarization and lack susceptible atoms to nucleophilic attacks by the described flavin intermediates of group C FPMOs. It has been suggested that LadA could operate through an alternative mechanism driven by a radical flavin intermediate, $Fl_{N5O}$ (Fig 3C). This path would lead to the activation and functionalization of the C-H bonds via a radical pathway -mimicking the mechanism of iron-binding enzymes like AlkB or CYP450 [32,38]. However, this alternative mechanism remains hypothetical, as our in vitro assays provided no evidence to support it. Furthermore, the reinterpretation of AlmA showing BVMO activity on aliphatic ketones rather than long-chain n-alkane monooxygenation and the results presented in this study suggest that metal-free group B and C FPMOs could be incapable of oxidizing n-alkanes. These findings thus raise critical questions about the role of FPMOs in n-alkane monooxygenation and whether metal-dependent enzymes (such as AlkB) are exclusively responsible for this enzymatic activity.

## Conclusions

Our comprehensive investigation into LadA's proposed role as a long-chain n-alkane monooxygenase revealed several unexpected insights that challenge our understanding of its previously proposed function. Although three prior studies from the same

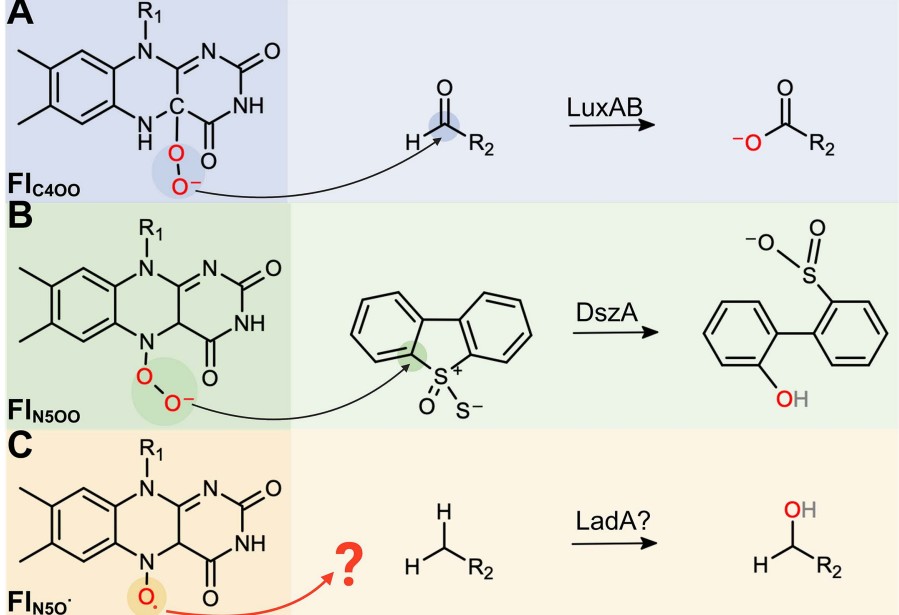

**Fig 3. Exemplary reactions catalyzed by the flavin intermediates of group C FPMOs.** (A) Baeyer-Villiger monooxygenation of aliphatic alkyl aldehydes to carboxylic acids catalyzed by luciferase, LuxA and LuxB. (B) Oxidative cleavage of dibenzothiophene sulfone to 2'-hydroxybiphenyl-2-sulfinate catalyzed by DszA. (C) Hypothetical radical-based mechanism for the oxidation of n-alkanes by LadA. $R_1$-ribityl phosphate. $R_2$-aliphatic chain.

research group suggested that LadA could catalyze *n*-alkane monooxygenation, the efforts of the Noda-García and Teufel groups to replicate this activity under various optimized *in vitro* conditions were unsuccessful. We meticulously recreated and modified experimental parameters based on the available literature, yet no monooxygenation activity was observed under any of those. In our search for functional variants, we also explored four LadA sequence homologs and generated one variant with higher solubility. Despite these efforts, neither the engineered LadA variant nor homologous sequences showed measurable *n*-alkane monooxygenation activity. This lack of activity across diverse homologs supports the notion that LadA may not act as a direct *n*-alkane monooxygenase. This hypothesis is further supported by an analysis of the biochemical constraints of the flavin-dependent mechanisms of (group C) FPMOs, which typically target polarized bonds, unlike the highly inert C-H bonds of *n*-alkanes. As previous *in vivo* studies supported LadA's involvement in *n*-alkane metabolism, we surmised that LadA could instead act further downstream on already oxidized *n*-alkanes and/or sulfur-containing compounds. However, LadA was also inactive when such ketones or sulfur compounds were offered as substrates. Overall, our findings challenge the proposal that LadA, and (group C) FPMOs generally can function as *n*-alkane monooxygenases and provide new incentives to scrutinize their exact roles.

## Materials and methods

### LadA, sequence homologs and DszA expression and purification

The nucleotide sequences of LadA, the PROSS designs, selected LadA homologs and DszA were codon-optimized with custom python scripts for expression in *E. coli*. The sequence of these proteins can be found in supplementary information, S1 Table. The optimized sequences were synthesized, cloned into the expression vector pET-28a(+) with an N-terminal StepTag added for affinity purification, and sequenced by Twist Biosciences (San Francisco, CA, USA). The constructs were transformed into One Shot™ BL21 Star™ (DE3) chemically competent *E. coli* cells (Thermo Fisher Scientific), plated on LB-Kan agar (kanamycin 50 mg/mL), and incubated overnight at 37°C.

A positive transformant colony was picked and grown overnight in 5 mL of LB-Kan to an optical density (OD) of 1.0-1.2, then used to inoculate a 1000-mL flask containing 500 mL of LB-Kan to an OD of 0.05. The culture was incubated at 37°C until the OD reached 0.4-0.6, at which point protein overexpression was induced with isopropyl β-D-1-thiogalactopyranoside (IPTG) to a final concentration of 0.5 mM. The culture was further incubated at 30°C for approximately 16 hours, except for DszA which was incubated at 15°C for 16 hours. Cells were harvested by centrifugation at 15,400 x g for 20 minutes at 4°C. The supernatant was discarded.

The cell pellet was resuspended in 10 mL of 50 mM TRIS-HCl buffer (pH 7.5) supplemented with Benzonase® Nuclease (Merck Millipore, Cat# 70746), EZBlock™ Protease Inhibitor Cocktail (EDTA-Free) (BioVision), and chicken egg white lysozyme (Sigma-Aldrich) (1 mg/mL). The cells were lysed by incubating on ice for 20 minutes, followed by sonication for 7 minutes (5 seconds pulse on, 5 seconds off, 60% of maximum power). Cell debris was removed by centrifugation at 15,400 x g for 20 minutes at 4°C. The supernatant was loaded onto a gravity column containing streptavidin StrepTag® resin (IBA Lifesciences GmbH) for affinity chromatography. The column was washed with 6 column volumes of washing buffer (100 mM Tris-HCl, pH 8, 150 NaCl, 1 mM EDTA) and elution was performed in six steps with 0.5 column volume each of washing buffer supplemented with 25 mM desthiobiotin. The purity of the eluted fractions was evaluated by electrophoresis using a 12.5% polyacrylamide gel. Total protein content was estimated by the coomassie-blue quantification method, through a calibration curve using bovine serum albumin as a standard protein. The typical yield per 500 mL of culture was 1.0–1.5 mg for LadA, 3.0–4.0 mg for LadA:P5, 1.5–2.0 mg for DszA, 0.1–0.5 mg for WP_149412683.1, 0.5–1.0 mg for WP_174015212.1, 0.1–0.5 mg for WP_088236120.1 and 0.8–1.0 mg for WP_023806228.1.

### Fre expression and purification

Flavin reductase (NCBI ID: WP_000209826.1) was prepared as standard molecular biology methods [24,32]. Briefly, the DNA sequence for Fre was amplified by PCR using *E. coli* K12 genomic DNA as a template with oligonucleotide primers Fre_F and Fre_R (S3 Table). The backbone was amplified by PCR from an empty pET28a(+) vector containing a StrepTag on the N-terminal end with primers pET28_F and pET28_R (S3 Table). The reactions were performed using Q5

polymerase according to the manufacturer's instructions (New England Biolabs). Then, the two amplified fragments were purified with QIAquick PCR Purification Kit (QIAGEN Labs, US) and assembled into a single construct with NEBuilder HiFi DNA Assembly reaction as indicated by the manufacturer (New England Biolabs). The construct was transformed into NEB 5-alpha competent *E. coli*, a derivative of DH5α (New England Biolabs,), plated on LB-Kan agar (kanamycin 50 mg/mL), and incubated overnight at 37°C. The functional plasmids were extracted with QIAprep Spin Miniprep Kit (QIAGEN Labs) from the overnight culture of the positive colonies and further extracted for sequence corroboration. The whole-plasmid sequencing was performed by Plasmidsaurus using Oxford Nanopore Technology with custom analysis and annotation. Protein expression and purification was carried out as earlier stated. Modifications to the protocol were the induction conditions, where IPTG was added to a final concentration of 0.25 mM, and cultures were incubated at 16°C for 4 hours. A typical yield ranges from 0.5–1.0 mg of pure protein per 300 mL of culture.

### Enzymatic activity assessment with hexadecane, 2-hexadecanone and dibenzothiophene sulfone

Reactions were prepared by mixing purified LadA and homologs (1–18 μM) with purified Fre (0.3-6 μM) in 50 mM Tris-HCl buffer, pH 7.5, containing 1 mM FMN, 1 mM hexadecane or 0.5 mM 2-hexadecanone, and 0.5% (w/v) HITENOL-AR. The total reaction volume was  mL, and the mixture was contained in a closed 4-mL amber glass vial equipped with a magnetic stirrer. Reactions were initiated by adding NADH to a final concentration of 1 mM and incubated at either 37°C or 60°C, in a water bath with stirring at 200 rpm.

Reactions for dibenzothiophene sulfone monooxygenase were prepared by mixing purified LadA:P5 or DszA (50 μM) with purified Fre (0.2 μM) in 50 mM Tris-HCl buffer, pH 7.5, containing 1 mM FMN and 0.5 mM dibenzothiophene sulfone. The total reaction volume was 3 mL, and the mixture was contained in a closed 4-mL amber glass vial equipped with a magnetic stirrer. Reactions were initiated by adding NADH to a final concentration of 1 mM and kept at 37°C, in a water bath with stirring at 200 rpm.

Every reaction was run for at least two technical replicates using two independently purified protein batches.

### GC-MS analysis

The reactions were analyzed at two time points: the initial stage (time zero, $t_0$) and two hours after the addition of NADH to the reaction medium. One milliliter of reaction mixture was sampled, and hydrophobic compounds were extracted three times with hexane (3 × 1 mL). Pentadecane was used as an internal standard. The combined organic layers were dried at room temperature. The resulting residue was dissolved in 1 mL of hexane, and 1 μL of this solution was injected into a gas chromatography-mass spectrometry (GC-MS) system (Agilent 7890B GC system coupled with a 5977B MSD). The separation was achieved using an HP-5ms column (30 m × 0.25 mm i.d., 0.25 μm film thickness), with helium as the carrier gas at a flow rate of 1.0 mL/min. The oven temperature was programmed from 70°C (hold for 0.5 min) to 280°C at a rate of 10°C/min, with a final hold time of 3 minutes. Mass spectra were recorded in electron impact (EI) mode at 70 eV. Compound identification was carried out by comparing mass spectra with spectral libraries and confirmed by matching retention times with standards.

### HPLC analysis

The analysis of the reactions was performed by HPLC as previously described, using a Thermo Fisher Vanquish Core 700 bar HPLC system equipped with a diode array detector (DAD) and a Luna C18 column (2.1 × 150 mm, 2.2 μm) [39].

### LadA homologous sequences selection

To identify homologous sequences of LadA, we queried the NCBI RefSeq database (August 2021) using pBLAST. The search aimed to retrieve sequences with significant similarity to the LadA enzyme from *Geobacillus thermodenitrificans*

*NG80−2* and the three homologs from *G. thermoleovorans* B23. Retrieved sequences were filtered based on 70% sequence coverage and 50% identity thresholds to ensure the inclusion of relevant homologs.

The retrieved sequences were aligned using MAFFT with default parameters to ensure accurate multiple sequence alignment. A cladogram was subsequently generated using IQ-TREE, employing default settings. This approach allowed us to visually group homologous sequences into distinct clades for downstream analysis.

To select representative sequences from each clade, a consensus sequence approach was applied. First, a consensus sequence was constructed for all sequences within a clade using alignment data. This consensus sequence was then used as a query for a BLAST search against the sequences in the same clade. The sequence with the highest identity to the consensus was selected as the representative sequence for that clade.

## Supporting information

**S1 Fig. GC-MS analyses for LadA.** Assayed reaction conditions are as follows: (A) *Wild-type* N-Strep-tag® LadA (18 µM), hexadecane (1 mM, $t_R$ = 12.74 min), FMN (1 mM), NADH (1 mM), MgCl$_2$ (1 mM) and HITENOL-AR 0.5% (m/v) for 2 hours at 37°C. (B) N-Strep-tag® LadA:P5 (18 µM), hexadecane (1 mM, $t_R$ = 12.74 min), FMN (1 mM), NADH (1 mM) and HITENOL-AR 0.5% (m/v) for 2 hours at 37°C.
(TIF)

**S2 Fig. GC-MS analyses for LadA and LadA:P5 reactions.** Assayed reaction conditions are as follows: (A) *Wild-type* N-Strep-tag® LadA (18 µM), *wild-type* N-Strep-tag® Fre (6 µM), hexadecane (1 mM, $t_R$ = 12.74 min), FMN (1 mM), NADH (1 mM) and HITENOL-AR 0.5% (m/v) for 2 hours at 37°C. (B) N-Strep-tag® LadA:P5 (18 µM), *wild-type* N-Strep-tag® Fre (6 µM), hexadecane (1 mM, $t_R$ = 12.74 min), FMN (1 mM), NADH (1 mM) and HITENOL-AR 0.5% (m/v) for 2 hours at 37°C. (C) N-Strep-tag® LadA:P5 (18 µM), *wild-type* N-strepTag Fre (6 µM), hexadecane (1 mM, $t_R$ = 12.74 min), FMN (1 mM), NADH (1 mM) and HITENOL-AR 0.5% (m/v) for 2 hours at 60°C. Additional peak at tR = 11.54 corresponds to pentadecane (internal standard).
(TIF)

**S3 Fig. Expression and assessment of LadA homologs.** (A) SDS-PAGE for *Pseudomonas* sp. ANT H4 (WP_149412683.1) and *P. viridiflava* p8B7 (WP_122428611.1) expression and purification. Lane 1 and 2 purified fractions of *Pseudomonas* sp. ANT H4. Lane 3 – Crude Lysate for *Pseudomonas* sp. ANT H4. Lane 4 and 5 – purified fractions of *P. viridiflava* p8B7. Lane 6 – Crude Lysate for *P. viridiflava* p8B7. Lane 7 – Molecular weight marker. (B) SDS-PAGE for *Mesorhizobium* sp. (WP_023806228.1) and *P. viridiflava* KF4851 (WP_088236120.1) expression and purification. Lane 1 – Molecular weight marker. Lane 2 – Crude Lysate for *Mesorhizobium* sp. expression. Lane 3 and 4 – purified fractions of *Mesorhizobium* sp. Lane 5 – Crude Lysate for *P. viridiflava* KF4851 Lane 6 and 7 – purified fractions of *P. viridiflava* KF4851 (C) SDS-PAGE for *A. rhizogenes* (WP_174015212.1) expression and purification. Lane 1 – Molecular weight marker. Lane 2 – Crude Lysate for *A. rhizogenes* expression. Lane 3 and 4 purified fractions of *A. rhizogenes*.
(TIF)

**S4 Fig. GC-MS analyses for LadA homologs.** Assayed reaction conditions are as follows: (A) *Wild-type* N-Strep-tag® LadA homolog from *Pseudomonas* sp. ANT H4 (WP_149412683.1) (18 µM), *wild-type* N-Strep-tag® Fre (6 µM), hexadecane (1 mM, $t_R$ = 12.74 min), FMN (1 mM), NADH (1 mM) and HITENOL-AR 0.5% (m/v) for 2 hours at 37°C. (B) *Wild-type* N-Strep-tag® LadA homolog from *P. viridiflava* p8B7 (WP_122428611.1) (18 µM), *wild-type* N-strepTag Fre (6 µM), hexadecane (1 mM, $t_R$ = 12.74 min), FMN (1 mM), NADH (1 mM) and HITENOL-AR 0.5% (m/v) for 2 hours at 37°C. (C) *Wild-type* N-Strep-tag® LadA homolog from *P. A. rhizogenes* AF44 96 (WP_174015212.1) (18 µM), *wild-type* N-Strep-tag® Fre (6 µM), hexadecane (1 mM, $t_R$ = 12.74 min), FMN (1 mM), NADH (1 mM) and HITENOL-AR 0.5% (m/v) for 2 hours at 37°C. (D) *Wild-type* N-Strep-tag® LadA homolog from *P. viridiflava* KF485.1 (WP_088236120.1) (18 µM), *wild-type* N-Strep-tag®

Fre (6 µM), hexadecane (1 mM, $t_R$ = 12.74 min), FMN (1 mM), NADH (1 mM) and HITENOL-AR 0.5% (m/v) for 2 hours at 37°C. Additional peak at tR = 11.54 corresponds to pentadecane (internal standard).
(TIF)

**S5 Fig. GC-MS analyses for LadA as BVMO.** Assayed reaction conditions are as follows: (A) N-Strep-tag® LadA:P5 (1 µM), *wild-type* N-Strep-tag® Fre (1 µM), hexadecanone (1 mM, $t_R$ = 15.1 min), FMN (1 mM), NADH (1 mM) and HITENOL-AR 0.5% (m/v) for 2 hours at 37°C. (B) N-Strep-tag® LadA:P5 (2 µM), *wild-type* N-Strep-tag® Fre (1 µM), hexadecanone (1 mM, $t_R$ = 15.1 min), FMN (1 mM), NADH (1 mM) and HITENOL-AR 0.5% (m/v) for 2 hours at 37°C. (C) N-Strep-tag® LadA:P5 (5 µM), *wild-type* N-Strep-tag® Fre (1 µM), hexadecanone (1 mM, $t_R$ = 15.1 min), FMN (1 mM), NADH (1 mM) and HITENOL-AR 0.5% (m/v) for 2 hours at 37°C.
(TIF)

**S6 Fig. HPLC analysis for dibenzothiophene sulfone monooxygenase activity for LadA.** Assayed reaction conditions are as follows: N-Strep-tag® LadA:P5 (50 µM) or N-Strep-tag® DszA, *wild-type* N-Strep-tag® Fre (0.2 µM), dibenzothiophene sulfone (0.5 mM, $t_R$ = 20.5 min), FMN (1 mM), NADH (1 mM) for 2 hours at 37°C.
(TIF)

**S1 Table. LadA PROSS-optimized variant sequences.**
(PDF)

**S2 Table. LadA selected homologous sequences.**
(PDF)

**S3 Table. Primers used for cloning Fre into NStrepTag-pET28a (+) vector.**
(PDF)

**S1 Raw data. SDS-PAGE gel images for Fig 1 and S3 Fig .**
(PDF)

## Acknowledgments

We thank Ella Jacob and Rachel Blank (The Institute of Chemistry, The Hebrew University of Jerusalem, Israel) for their assistance with the GC-MS analyses.

## Author contributions

**Conceptualization:** Lianet Noda-García, Raul Mireles.

**Investigation:** Raul Mireles.

**Writing – original draft:** Lianet Noda-Garcia, Raul Mireles.

**Writing – review & editing:** Lianet Noda-Garcia, Raul Mireles, Arne Matthews, Robin Teufel.

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
