## [Decision Letter · Decision Letter 0]

13 Jun 2025

PONE-D-25-27398Can flavoprotein monooxygenases functionalize long-chain n-alkanes?PLOS ONE

Dear Dr. Noda-Garcia,

Thank you for submitting your manuscript to PLOS ONE. After careful consideration, we feel that it has merit but does not fully meet PLOS ONE’s publication criteria as it currently stands. Therefore, we invite you to submit a revised version of the manuscript that addresses the points raised during the review process.

We look forward to receiving your revised manuscript.

Kind regards,

Kshatresh Dutta Dubey

Academic Editor

PLOS ONE

Journal Requirements:

“Proposal no. 12120006

Ministry of Agriculture and Rural Development.”

4. We notice that your supplementary figures are uploaded with the file type 'Figure'. Please amend the file type to 'Supporting Information'. Please ensure that each Supporting Information file has a legend listed in the manuscript after the references list.

Reviewers' comments:

Reviewer's Responses to Questions

**Comments to the Author**

1. Is the manuscript technically sound, and do the data support the conclusions?

Reviewer #1: Yes

Reviewer #2: Yes

Reviewer #3: Yes

2. Has the statistical analysis been performed appropriately and rigorously? 

Reviewer #1: N/A

Reviewer #2: Yes

Reviewer #3: N/A

3. Have the authors made all data underlying the findings in their manuscript fully available?

Reviewer #1: Yes

Reviewer #2: Yes

Reviewer #3: Yes

4. Is the manuscript presented in an intelligible fashion and written in standard English?

Reviewer #1: Yes

Reviewer #2: Yes

Reviewer #3: Yes

5. Review Comments to the Author

Reviewer #1: This study reports on something that is often encountered, but is rarely published – a negative result. In this case, a fairly exhaustive attempt was made to replicate the prior, published observation that LadA has n-alkane monooxygenase activity in vitro. An array of assay conditions were attempted and multiple LadA homologs from other organisms were assessed. In all cases, no n-alkane monooxygenase activity was detected, suggesting that LadA-like enzymes are unable to catalyze the previously published reaction and the true function of LadA is still unknown. The fact that this was independently evaluated by two separate groups in this study (Noda-Garcia and Teufel groups) provides an additional level of confidence in the findings. This study appears to be well-conducted and provides information that will be useful for anyone studying LadA. However, I have one major suggesting for the authors before publishing this study:

1. Assess LadA for activity under conditions with MgSO4, with and without Fre, similar to the published studies, even though they do not seem mechanistically sound. I realize that there is no rationale for needing Mg2+ and a flavin reductase is usually necessary for these two component systems to function. In my experience, NADH can nonenzymatically reduce FMN, albeit poorly, so it is conceivable that FMNH2 could still be generated in the assay without Fre. I appreciate that the results of this additional experiment will very likely be the same as all of your others (no activity). Without it, though, a skeptical reader of your study may wonder if you didn’t observe activity with LadA simply because you opted not to use similar conditions to the the prior studies.

Minor suggestion:

2. Be a little more forceful in pointing out how nonsensical some of the conditions are from the prior published studies. “In the study by Feng et al. (2007), the reaction setup included 1 mM hexadecane and 1 mM MgSO₄ in a 50 mM Tris-HCl buffer at pH 7.5, along with 1 mM NADH.” It is unclear how LadA activity could be detected in those experiments without FMN added since Group C FMOs usually do not purify with FMN bound. “The second study by Li et al. (2008), which focused on the structural analysis of LadA, also used 1 mM hexadecane in a 50 mM Tris-HCl buffer at pH 7.5; however, NADH was omitted, and 1 mM FMNH2 was added instead.” No information was provided to describe how the FMNH2 was generated for these experiments. This is critical information, as FMNH2 is not stable under aerobic conditions.

Reviewer #2: This manuscript addresses the challenging issue of n-alkane functionalization. The authors have undertaken an effort to clarify the enzyme activity of the flavin-dependent enzyme LadA, an enzyme previously suggested to function as a long-chain n-alkane monooxygenase. Their rigorous experimentation, including testing optimized variants and novel homologs under in vitro conditions, significantly advances the current understanding.

Our research group has also been particularly intrigued by LadA and pursued extensive studies aimed at reproducing the previously reported literature findings, unfortunately without success. Therefore, we highly appreciate the authors' systematic approach in clearly demonstrating and openly discussing the reproducibility challenges and limitations related to LadA activity. Their results provide crucial clarity and offer a balanced and critical evaluation of the previously reported roles of LadA and related flavoprotein monooxygenases.

I congratulate the authors on this valuable and rigorous study, which strongly contributes to redefining the biochemical understanding of these enzymes. Their findings represent an essential milestone and open new avenues for research to uncover the true biochemical roles of LadA and its homologs.

Reviewer #3: Dear Authors

that is a very important finding and clarifies many open questions we had for this special FPMO, it also demonstrates how experimental setup can be designed to answer research questions from different perspectives. I have some minor mostly text-related suggestions which I hope help to improve the manuscript.

Comments:

- “sp.“ should read italics (e.g., page 3 “AlmA in Acinetobacter …”

- “E. coli” should read italics

- “(18 μM) LadA to our assays,,” comma issue to be corrected

- Be consistent with “n-alkane” in terms of “n” reads italics

- Protein expression is wrong; only genes can be expressed; correct throughout.

- In your assays you provide Fre to provide reduced flavin for the potential monooxygenase; did you observe any H2O2 formation and if; was it different plus/minus substrate being present, that would at least indicate a ligand binding and thus effector role.

- Fre might not be the best reductase partner as some two-component FPMOs need a certain partner to become active; like cooperativity and this might be an important conclusion to search for the natural redox couple.

- If the potential uncoupling was high and the redFMN transfer not efficient, the 1 mM NADH might be too little to see product formation or only traces. However, I think in your setup you should have seen a little bit of product – so not point to repeat.

6. PLOS authors have the option to publish the peer review history of their article (what does this mean? ). If published, this will include your full peer review and any attached files.

**Do you want your identity to be public for this peer review?** For information about this choice, including consent withdrawal, please see our Privacy Policy .

Reviewer #1: No

Reviewer #2: **Yes: ** Frank Hollmann

Reviewer #3: **Yes: ** Dirk Tischler

---

## [Author Response · Author response to Decision Letter 1]

20 Jul 2025

Review Comments to the Author

Reviewer #1

This study reports on something that is often encountered, but is rarely published – a negative result. In this case, a fairly exhaustive attempt was made to replicate the prior, published observation that LadA has n-alkane monooxygenase activity in vitro. An array of assay conditions were attempted and multiple LadA homologs from other organisms were assessed. In all cases, no n-alkane monooxygenase activity was detected, suggesting that LadA-like enzymes are unable to catalyze the previously published reaction and the true function of LadA is still unknown. The fact that this was independently evaluated by two separate groups in this study (Noda-Garcia and Teufel groups) provides an additional level of confidence in the findings. This study appears to be well-conducted and provides information that will be useful for anyone studying LadA. However, I have one major suggesting for the authors before publishing this study:

1. Assess LadA for activity under conditions with MgSO4, with and without Fre, similar to the published studies, even though they do not seem mechanistically sound. I realize that there is no rationale for needing Mg2+ and a flavin reductase is usually necessary for these two component systems to function. In my experience, NADH can nonenzymatically reduce FMN, albeit poorly, so it is conceivable that FMNH2 could still be generated in the assay without Fre. I appreciate that the results of this additional experiment will very likely be the same as all of your others (no activity). Without it, though, a skeptical reader of your study may wonder if you didn’t observe activity with LadA simply because you opted not to use similar conditions to the prior studies.

Thank you for your comment. Indeed, the absence of identical conditions might raise questions. We performed the experiments as originally reported by adding Mg2+ with and without Fre. As expected, we did not observe any hexadecane conversion or hexadecanol formation. These results are now explicitly referenced in the main text, and the chromatograms are presented as Supplementary Figure 1.

Also, due to this new experiment the test now read as:

“In the study by Feng et al. (2007), the reaction setup included 1 mM hexadecane and 1 mM MgSO₄ in a 50 mM Tris-HCl buffer at pH 7.5, along with 1 mM NADH. It is unclear how LadA activity could be detected in those experiments without FMN added, since Group C FPMOs usually do not purify with FMN bound. The second study by Li et al. (2008), which focused on the structural analysis of LadA, also used 1 mM hexadecane in a 50 mM Tris-HCl buffer at pH 7.5; however, NADH was omitted, and 1 mM FMNH2 was added instead. No information was provided to describe how the FMNH2 was generated for these experiments. This is critical information as it is unstable under aerobic conditions. ”

Minor suggestion:

2. Be a little more forceful in pointing out how nonsensical some of the conditions are from the prior published studies. “In the study by Feng et al. (2007), the reaction setup included 1 mM hexadecane and 1 mM MgSO₄ in a 50 mM Tris-HCl buffer at pH 7.5, along with 1 mM NADH.” It is unclear how LadA activity could be detected in those experiments without FMN added since Group C FMOs usually do not purify with FMN bound. “The second study by Li et al. (2008), which focused on the structural analysis of LadA, also used 1 mM hexadecane in a 50 mM Tris-HCl buffer at pH 7.5; however, NADH was omitted, and 1 mM FMNH2 was added instead.” No information was provided to describe how the FMNH2 was generated for these experiments. This is critical information, as FMNH2 is not stable under aerobic conditions.

Thanks for the observations. The text was corrected, and the three new sentences suggested were added.

Reviewer #2

This manuscript addresses the challenging issue of n-alkane functionalization. The authors have undertaken an effort to clarify the enzyme activity of the flavin-dependent enzyme LadA, an enzyme previously suggested to function as a long-chain n-alkane monooxygenase. Their rigorous experimentation, including testing optimized variants and novel homologs under in vitro conditions, significantly advances the current understanding.

Our research group has also been particularly intrigued by LadA and pursued extensive studies aimed at reproducing the previously reported literature findings, unfortunately without success. Therefore, we highly appreciate the authors' systematic approach in clearly demonstrating and openly discussing the reproducibility challenges and limitations related to LadA activity. Their results provide crucial clarity and offer a balanced and critical evaluation of the previously reported roles of LadA and related flavoprotein monooxygenases.

I congratulate the authors on this valuable and rigorous study, which strongly contributes to redefining the biochemical understanding of these enzymes. Their findings represent an essential milestone and open new avenues for research to uncover the true biochemical roles of LadA and its homologs.

Thank you for your kind words. We agree that our attempts in investigating LadA’s activity may ultimately contribute to uncovering its native function and advance the broader understanding of FPMOs and enzymatic C–H functionalization.

Reviewer #3

Dear Authors

that is a very important finding and clarifies many open questions we had for this special FPMO, it also demonstrates how experimental setup can be designed to answer research questions from different perspectives. I have some minor mostly text-related suggestions which I hope help to improve the manuscript.

Thank you very much for your thoughtful feedback. We're glad to hear that the findings help clarify open questions regarding this unusual FPMO. We fully agree that the design of the experimental setup was key to approaching the problem from multiple angles. We appreciate your text-related suggestions and will incorporate them to further improve the clarity and quality of the manuscript.

Comments:

- „sp.“ should read italics (e.g., page 3 “AlmA in Acinetobacter …”

Corrected.

- “E. coli” should read italics

Corrected.

- “(18 μM) LadA to our assays,,” comma issue to be corrected

Corrected.

- Be consistent with “n-alkane” in terms of “n” reads italics

Corrected. “n-alkane” is consistent throughout the text.

- Protein expression is wrong; only genes can be expressed; correct throughout.

Corrected. The wording ”protein expression” and variants were changed to “protein production” throughout the text.

- In your assays you provide Fre to provide reduced flavin for the potential monooxygenase; did you observe any H2O2 formation and if; was a different plus/minus substrate being present, that would at least indicate a ligand binding and thus effector role.

We did not measure H2O2 formation.

- Fre might not be the best reductase partner as some two-component FPMOs need a certain partner to become active; like cooperativity and this might be an important conclusion to search for the natural redox couple.

This is an interesting observation. Our decision to use a previously characterized flavin reductase was based on the fact that ladA is most likely acquired through horizontal gene transfer—being encoded on a plasmid in the native host—while no flavin reductase gene is present on the same plasmid. This supports the hypothesis that LadA may interact promiscuously with host-encoded flavin reductases.

Group C FPMOs have generally been shown to function promiscuously with reductase partners. See references

Su, T., 2018: https://doi.org/10.3389/fmicb.2018.00231

Matthews A., 2020: https://doi.org/10.1038/s41589-020-0476-2

Mathews, A., et al., 2021: https://doi.org/10.1111/febs.16193

Moreover, the transfer of reduced flavin from the reductase to the monooxygenase component has primarily been proposed to occur via free diffusion. A comprehensive review on this topic is provided by Sucharitakul et al. (2014) https://doi.org/10.1016/j.abb.2014.05.009.

- If the potential uncoupling was high and the redFMN transfer not efficient, the 1 mM NADH might be too little to see product formation or only traces. However, I think in your setup you should have seen a little bit of product – so not point to repeat.

Thank you. Indeed, we hypothesized that an excess of NADH would be sufficient to observe at least trace amounts of hexadecane conversion.

---

## [Decision Letter · Decision Letter 1]

4 Sep 2025

Can flavoprotein monooxygenases functionalize long-chain n-alkanes?

PONE-D-25-27398R1

Dear Dr. Noda-Garcia,

We’re pleased to inform you that your manuscript has been judged scientifically suitable for publication and will be formally accepted for publication once it meets all outstanding technical requirements.

Kind regards,

Paul Aurelian Gagniuc, PhD

Academic Editor

PLOS ONE

Reviewers' comments:

Reviewer's Responses to Questions

**Comments to the Author**

1. If the authors have adequately addressed your comments raised in a previous round of review and you feel that this manuscript is now acceptable for publication, you may indicate that here to bypass the “Comments to the Author” section, enter your conflict of interest statement in the “Confidential to Editor” section, and submit your "Accept" recommendation.

Reviewer #1: All comments have been addressed

Reviewer #2: All comments have been addressed

Reviewer #3: All comments have been addressed

2. Is the manuscript technically sound, and do the data support the conclusions?

Reviewer #1: Yes

Reviewer #2: Yes

Reviewer #3: Yes

3. Has the statistical analysis been performed appropriately and rigorously? 

Reviewer #1: Yes

Reviewer #2: Yes

Reviewer #3: Yes

4. Have the authors made all data underlying the findings in their manuscript fully available?

Reviewer #1: Yes

Reviewer #2: Yes

Reviewer #3: Yes

5. Is the manuscript presented in an intelligible fashion and written in standard English?

Reviewer #1: Yes

Reviewer #2: Yes

Reviewer #3: Yes

6. Review Comments to the Author

Reviewer #1: (No Response)

Reviewer #2: A very important paper. Ich have no further objections and I am a bit disturbed that my answer needs to be at least 100 characters.

Reviewer #3: All previous comments have been addressed. Well done! No further changes are needed from point of view.

7. PLOS authors have the option to publish the peer review history of their article (what does this mean? ). If published, this will include your full peer review and any attached files.

**Do you want your identity to be public for this peer review?** For information about this choice, including consent withdrawal, please see our Privacy Policy .

Reviewer #1: No

Reviewer #2: **Yes: ** Frank Hollmann

Reviewer #3: No

---

## [Editor Report · Acceptance letter]

PONE-D-25-27398R1

PLOS ONE

Dear Dr. Noda-Garcia,

I'm pleased to inform you that your manuscript has been deemed suitable for publication in PLOS ONE. Congratulations! Your manuscript is now being handed over to our production team.

Kind regards,

on behalf of

Dr. Paul Aurelian Gagniuc

Academic Editor

PLOS ONE